# Efficient Deep Wavelet Gaussian Markov Dempster–Shafer Network-Based Spectrum Sensing at Very Low SNR in Cognitive Radio Networks

**DOI:** 10.3390/s25237361

**Published:** 2025-12-03

**Authors:** Sunil Jatti, Anshul Tyagi

**Affiliations:** Department of Electronics and Communication Engineering, Indian Institute of Technology Roorkee, Roorkee 247667, India; jmadhukarrao@ec.iitr.ac.in

**Keywords:** cognitive radio network, spectral sensing, deep learning, primary user activity detection, reinforcement learning for spectrum sensing, low SNR

## Abstract

Cognitive radio networks (CRNs) rely heavily on spectral sensing to detect primary user (PU) activity, yet detection at low signal-to-noise ratios (SNRs) remains a major challenge. Hence, a novel “Deep Wavelet Cyclostationary Independent Gaussian Markov Fourier Transform Dempster–Shafer Network” is proposed. When the signal waveform is submerged within the noise envelope and residual correlation emerges in the noise, it violates white Gaussian assumptions, leading to misidentification of signal presence. To resolve this, the Adaptive Continuous Wavelet Cyclostationary Denoising Autoencoder (ACWC-DAE) is introduced, in which the Adaptive Continuous Wavelet Transform (ACWT), Cyclostationary Independent Component Analysis Detection (CICAD), and Denoising Autoencoder (DAE) are introduced into the first hidden layer of a Deep Q-Network (DQN). It restores the bursty signal structure, separates the structured noise, and reconstructs clean signals, leading to accurate signal detection. Additionally, bursty and fading-affected primary user signals become fragmented and dip below the noise floor, causing conventional fixed-window sensing to fail in accumulating reliable evidence for detection under intermittent and low-duty-cycle conditions. Therefore, the Adaptive Gaussian Short-Time Fourier Transform Dempster–Shafer Model (AGSTFT-DSM) is incorporated into the second DQN layer, Adaptive Gaussian Mixture Hidden Markov Modeling (AGMHMM) tracks the hidden activity states, Adaptive Short-Time Fourier Transform (ASFT) resolves brief signal bursts, and Dempster–Shafer Theory (DST) fuses uncertain evidence to infer occupancy, thereby detecting an accurate user signal. The results obtained by the proposed model have a low error and detection time of 0.12 and 30.10 ms and a high accuracy of 97.8%, revealing the novel insight that adaptive wavelet denoising, along with uncertainty-aware evidence fusion, supports reliable spectrum detection under low-SNR conditions where existing models fail.

## 1. Introduction

In wireless communication, cognitive radio networks (CRNs) offer a promising area of research for dynamic spectrum access to overcome spectrum scarcity challenges. In a CRN, spectrum sensing is a subarea that enables secondary users to evaluate the presence or absence of primary users on a particular frequency band. The advantage of CRNs is the added possibility of improving spectrum use without causing interference to licensed users. CRNs face many challenges, particularly in non-ideal channel conditions. For example, conventional energy detection-based methods have poor performance in fading channels. Cyclostationary detection can perform better; however, it can also consume high computational resources and have increased observation time. Matched filtering has optimal performance; however, it requires prior knowledge of primary user (PU) characteristics, which is often not possible. The above restrictions necessitate the development of more adaptive and intelligent sensing techniques within the context of dynamic and noisy spectral environments [1,2,3,4].

Several techniques of sensing spectrum have been utilized to mitigate the problem of deciding whether a primary user (PU) is using a frequency band so that opportunistic access by a secondary user (SU) can be facilitated. Energy detection (ED) is the most widely used technique due to its ease of implementation, but it has lower performance at low SNR and is very sensitive to noise uncertainty. Cyclostationary Feature Detection (CFD) takes advantage of the periodicity of modulated PU signals and provides more robustness against noise but at the cost of high computational and observation time expenses, thus being inappropriate for rapid sensing. Matched filtering operates best but is disadvantaged by the requirement of a priori information of the PU signal, something that does not happen in most dynamic CRN scenarios. Eigenvalue-based methods like Maximum–Minimum Eigenvalue (MME) and Energy With Minimum Eigenvalue (EME) try to deal with correlated noise conditions for sensing but are computationally costly in terms of large samples and intricate operations. Wavelet-based sensing methods permit spectral edge or transition detection but are very dependent on wavelet and scale resolution. Compressive Sensing (CS) compresses sensing overhead with signal sparsity but is prone to noisy environments due to recovery errors. In all of these methods, the SU is confronted with being able to reliably detect the presence of the PU in a band of spectrum, especially when the PU signal is bursty, low in power, or degraded. Detection of low-power or intermittent PU activity is still one of the unsolved problems of efficient spectrum use and PU protection [5,6,7,8].

To enhance detection capabilities, many artificial intelligence (AI) techniques have incorporated spectrum sensing. Support Vector Machines (SVMs) and k-Nearest Neighbors (k-NNs) have been used to classify received signal feature sets but typically require significant nonlinear training, as well as being sensitive to the quality of the selected features. Other models, including neural networks (feedforward and convolutional), have shown potential for learning nonlinear representations of signals but tend to often overfit, especially in noisy environments. In addition, there has been the use of recurrent neural networks (RNNs) as well as Long Short-Term Memory (LSTM) models to capture temporal signal dependencies; however, performance degrades considerably when noise is excessive or when data is missing. Deep learning models, including CNN, have the function of providing some robustness in terms of automatic feature extraction, but they also require large datasets and substantial computational power. Finally, reinforcement learning techniques have been proposed to develop learned policies from the domain of exploration, increasing the likelihood of optimization; however, they are slow to converge and thus highly dependent on the states selected for exploration as a result of low generalization ability. Despite all of these advances in AI, challenges for generalizability exist in low-resource highly dynamic spectrum environments [9,10,11,12].

Under extremely low-SNR conditions, conventional spectrum sensing cannot be reliably executed. Energy detection breaks down due to noise uncertainty and false alarms. Cyclostationary-based methods are theoretically sub-band noise-resistant; however, these methods do not detect weak signals below the noise floor unless the sensing duration is rather long. Eigenvalues and entropy rely on large sample covariance matrices, which tend to be very unstable for low-SNR scenarios. Although there has been discussion of using wavelet transforms and multiscale analysis for the localization of the weak signal features, again, under low SNR, they are sensitive to the scale and shift parameters, resulting in ever-increasing errors in predictions. Matched filtering assumes the availability of known PU signal templates, rendering it impossible under very low SNR. Even compressed sensing will have difficulty reconstructing sparse signals from noisy observations. Recent attempts to use deep learning under very low SNR have shown some improvement, but the neural net is limited in the quality of the descriptor input features, and they are unable to accommodate working with incomplete or fragmented observations of the original signal. Moreover, spectrum sensing for PU users in very low-SNR conditions remains unsolvable due to the inherent effects of noise, fading, and signal fragmentation [13,14,15].

Despite significant progress in spectrum sensing for cognitive radio networks, the existing techniques still face critical limitations in reliably detecting primary users under challenging conditions. Conventional methods often fail to maintain accuracy when signals are weak, intermittent, or distorted by noise and fading. Key issues such as signal submergence, chaotic autocorrelation behavior, and spectral leakage continue to hinder sensing reliability. Additionally, most models lack robustness in distinguishing structured interference from true PU activity. These limitations affect the efficiency and safety of dynamic spectrum access. Therefore, new solutions are needed to improve detection accuracy and decision confidence in low-SNR and spectrally crowded environments.

### 1.1. Contributions of the Research

This research addresses the new and serious research gap in spectrum sensing under low-SNR conditions, where the existing methods struggle to differentiate primary user activity from structured noise. The main contributions of the research are as follows:This research proposed the Adaptive Continuous Wavelet Cyclostationary Denoising Autoencoder (ACWC-DAE), which combines the Adaptive Continuous Wavelet Transform, Cyclostationary Independent Component Analysis Detection, and Denoising Autoencoder to suppress chaotic autocorrelation, recover temporal structure, and denoise signals under low SNR. This method effectively mitigates the dual degradation modes caused by signal submergence and structured noise interference.The second contribution of this research is the Adaptive Gaussian Short-Time Fourier Transform Dempster–Shafer Model (AGSTFT-DSM), which integrates Adaptive Gaussian Mixture Hidden Markov Modeling, Adaptive Short-Time Fourier Transform, and Dempster–Shafer Theory. This proposed model handles bursty and fading-affected signal transitions, adaptively resolves brief temporal segments, and performs uncertainty-aware decision fusion, supporting accurate detection in deep fading and low-duty-cycle scenarios.By embedding ACWC-DAE in the first layer and AGSTFT-DSM in the second layer of the Deep Quality Network (DQN), this proposed system is capable of learning resilient sensing policies from a noisy environment, reducing dependence on large labeled datasets. The final output layer of the DQN performs binary classification to determine the presence or absence of the primary user, even under highly degraded low-SNR environments.

From the above contributions, an adaptive, noise-resilient, and uncertainty-aware deep learning model for cognitive radio networks is proposed, addressing the persistent issues of accurately detecting the primary user under low-SNR conditions.

This research exposes that the main reason for the failure of existing low-SNR spectrum sensing is due to two degradation modes: first, the submergence of weak signal structure within noise; second, the increase in structured noise correlation that disturbs Gaussian assumptions. From the proposed dual-layer Deep Q-Network framework, a new integration of methods is developed between adaptive wavelet denoising and probabilistic evidence fusion that is able to restore temporal coherence before uncertainty reasoning, thus improving sensing reliability. The model achieves high detection accuracy even at low SNR, which validates its capability of extracting significant signal evidence in environments where existing energy, eigenvalue, or cyclostationary methods fail.

### 1.2. Organization of the Paper

The previously discussed contributions aim to overcome the core limitations of conventional spectrum-sensing approaches by integrating adaptive time–frequency analysis, cyclostationary detection, probabilistic modeling, and uncertainty-aware decision fusion within a deep reinforcement learning framework. The structure of this research is as follows: Section 2 presents a detailed review of existing spectrum-sensing techniques, with an emphasis on autocorrelation-based methods, time–frequency transforms, and AI-based detection under low-SNR conditions. Section 3 describes the proposed Deep Wavelet Cyclostationary Independent Gaussian Markov Fourier Transform Dempster–Shafer Network. Section 4 provides performance evaluation and comparative analysis with existing spectrum-sensing methods in various low-SNR cognitive radio scenarios. Finally, Section 5 concludes the study by summarizing the key findings and highlighting potential directions for future work in adaptive and learning-based spectrum sensing.

## 2. Literature Survey

Sara E. Abdelbaset et al. [16] developed a new spectrum-sensing approach that increased accuracy and efficiency in identifying unused frequency bands by leveraging the use of CNN. The model was assigned different types of signal and noise datasets, and the problem of spectrum sensing was framed as a classification task to attain unparalleled capability to cope with new signals. The method executed and revised better than traditional methods, such as eigenvalue ratio and frequency domain-based methods. Particularly, the CNN-based method performed extremely well, even outperforming earlier methods in AWGN scenarios. The model performed weaker in regard to generalization ability when faced with actual non-Gaussian patterns of noise outside the learning dataset.

Vinodkumar Mohanakurup et al. [17] presented DRAIN-NETS for enhancing spectrum-sensing performance. The network was proposed to learn temporal characteristics from spectral information and take advantage of environmental activity statistics like energy, distance, and duty cycle time to enhance sensing. The system was evaluated on diverse platforms; the introduced approach was based on thresholding under certain signal–noise model assumptions, so its detection performance was strongly sensor accuracy-sensitive. Further, DRAIN-NETS necessitated fine-grained calibrations for varying hardware configurations, making portability across heterogeneous devices restricted.

Surendra Solanki et al. [18] proposed DLSenseNet, which utilized the structural data of received controlled signals for spectrum sensing. The model was evaluated using a diverse dataset, and the experimental results demonstrated that DLSenseNet achieved better spectrum-detection performance than other models. The architecture was enhanced by adapting the inception module and incorporating LSTM and fully connected layers to capture temporal dependencies and spatial relations. Nevertheless, the increased model complexity led to higher inference latency, which may not suit real-time spectrum-sensing applications.

Chen Wang et al. [19] proposed a new adversarial learning-based technique for spectrum sensing to increase model robustness. The principle of the technique was that three coupled neural networks were created to extract the universal features, which were less dependent on SNR, from the training SNR set to infer the status of the spectrum in test SNR sets. The simulation results showed a marked improvement in spectrum sensing error rate compared to existing machine learning and conventional signal-processing methods. However, the adversarial training did have performance inconsistency under varying initialization conditions, suggesting that a more generalized and consistent adversarial learning methodology is a research gap to pursue.

Ajayi et al. [20] developed a deep learning-based detection (DLbD) method that could overcome some limitations of non-cooperative devices operating in low-SNR environments in CRNs. Their method used an LSTM model to extract features from modulated received signals, allowing the LSTM classifier to accurately distinguish between signals and noise in a spectrum band. Training of the model made use of simulated signals and randomly generated data of different modulation schemes and noise. The performance metrics for DLbD compared to ED and CFD included probability of detection and probability of missing. DLbD indicated strong potential for integration in 5G ultra-dense networks and future use in intelligent systems within smart city contexts. However, one of the challenges in deploying their scheme in a real-world setting is that the training process requires large amounts of labeled data, which can be challenging to obtain.

Sanjeevkumar Jeevangi et al. [21] introduced a multi-stage detector for strong signal/spectra-sensing applications. In the first stage, the sample signal was evaluated for SNR estimation using a CNN. Then, the detection strategy was chosen according to the predicted SNR levels of the received signal. ED and SVD were the solutions used for high SNR, and refined MNMF was the solution used for low SNR. The CNN weights were chosen for the neural network models generated using LU-SLNO, an updated method of the traditional SLNO method. However, by relying on preset SNR thresholds, the model put itself at risk of misled SNR predictions, compromising the selected detector method.

Yogesh Mishra et al. [22] proposed a DL approach to automatically categorize received raw signal data, which was thought to be time-series data. The results were provided using different trial lengths and SNR, low or high. The performance of the ResNet model had the highest detection probabilities in both low- and high-SNR ratios. The simulation results of the proposed method improved the system accuracy and decreased the loss for false alarms in prediction while increasing detection probability. Furthermore, it only considered ResNet, which limited the discovery (or consideration) of hybrid architectures that could have improved interpretability and efficiency.

Vargil Vijay et al. [23] introduced a CNN–TN that combined CNNs and TNs to make the best use of the available spectrum. The CNN was beneficial for capturing features in the signals, helping with the feature selection in the IQ signal. The transformer accounted for long-range temporal dependencies and improved the selected features. This method was able to reduce sensing errors and increase the probability of detection by using both local and global features of the spectrum data. The performance metrics used to evaluate the method were Cohen’s Kappa coefficient and F1 score, which outperformed previous methods. However, the model also had high computational costs since both the CNN and transformer networks are expensive and using both in a single model raises complexities for real-time application.

Surendra Solanki et al. [24] presented a CNN–RNN, which had CNN and RNN layers. The model was built as a CNN–RNN design in order to indirectly learn significant features in the spatio-temporal spectrum data. The CNN was used to perform feature extraction, and the RNN was used to exploit the temporal characteristics in the spectrum data. The ideal model had two layers in parallel, and the output of those layers was then passed through a dropout layer to provide effective regularization and improved generalization performance. The output was then concatenated and passed through a flatten layer in order to arrange the dimensions for the next layers. The parallel convolutional structure increased memory overhead, thus increasing difficulty regarding on-edge resource-constrained assets.

Liuwen Li et al. [25] designed an obliging spectrum-sensing solution based on the parallel connection of a CNN and an LSTM, exploiting the complementary characteristics of both networks for feature extraction, combining the hidden spatial features from the CNN and the temporal features from the LSTM. The original dataset was a direct input to the model due to the parallel connection of the two networks, which prevents the loss of feature information that occurs when the connection is serial. The experimental results demonstrated that the algorithm achieved better detection performance compared with traditional spectrum detection algorithms under low-SNR conditions; however, the parallel connection in the model introduced synchronization issues during training that could disrupt the stability of convergence.

Saraswathi et al. [26] introduced a novel spectral-sensing technique for cognitive radio networks (SST–CRN), which reduced the limitations of energy-detection models. The DBN was included in the introduced model to achieve the nonlinear threshold based on the chicken swarm algorithm (CSA). This DBN was enabled for the SST–CRN approach, which was divided into two phases: (a) online and (b) offline. In the offline phase, the pre-collected data was used to train the DBN to smartly distinguish the problematic patterns and the examples from the spectral features of the CRN environment. Moreover, online spectrum sensing was performed during the actual communication phase to allow real-time adaptation to the dynamic changes in the spectrum environment. The proposed solution tapped the DBN’s potential to considerably improve the spectrum efficiency and resilience of CRNs, thus leading to effective resource utilization and less interference. However, the model’s non-scalability in large-scale dynamic CRN environments significantly hindered its real-world deployment potential and was thus viewed as a major limitation.

Manpreet Kaur and others [27] applied the supervised ML algorithm SVM for primary user detection and examined the various kinds of SVMs, like linear, polynomial, and Gaussian radial function, in addition to incorporating the ensemble classification-based method to boost the classifier’s output and performance. Each secondary user was allowed to conduct energy detection-based spectrum sensing locally in a distributed manner as part of the proposed approach. The results generated through simulation indicated that the ensemble classifier performed best, followed by the RBF SVM classifier, based on the performance metrics AUC and ROC. However, better performance was accompanied by higher computational complexity. One of the major limitations was that the model’s accuracy reduced drastically under low-SNR conditions, which in turn hindered reliable primary user detection.

Dhivya [28] proposed a new framework that relies on both DRL and CNNs, which allowed for more context-sensitive and adaptive spectrum sensing in IoT-enabled CRNs. The first part of the framework consisted of a CNN model that was supposed to extract features from the spectrogram of the radio frequencies and thus provide a representation of the signals’ varying density. The features of the RF were then given to the DRL agent, which in turn enhanced its sensing policy by interacting with a simulated network considering wireless parameters like SNR, energy status, movement speed, and noise levels. The hybrid model was then tested in different IoT scenarios, such as stationary and mobile nodes, various SNRs, and heavy interference, and the performance of each case was tracked. Moreover, the suggested framework could not perform well without a large amount of training data and powerful computing resources, making adaptation in dynamic IoT networks take longer in real time.

Aman Kumar et al. [29] suggested a spectrum-sensing technique that utilized the Fast Slepian Transform (FST) and took advantage of the time–frequency concentration properties of Discrete Prolate Spheroidal Sequences (DPSSs) to map received signals to a sparse spectral domain. Thus, the occupied sub-bands were detected efficiently. The fixed and adaptive thresholding techniques were tested for their performances, where the adaptive threshold for detection was optimized using the F1 score to achieve a good compromise between the two parameters, such as detection accuracy and false alarms. Moreover, the computational efficiency metric is based on the number of floating-point operations (FLOPs) to determine the energy cost of detection. However, the extreme sensitivity of the FST approach to the selection of the bandwidth parameter in DPSS, and even a slight misconfiguration, led to a considerable drop in sub-band localization accuracy.

Kai Wang et al. [30] tried to improve spectrum sensing in multi-user cooperating cognitive radio systems by using a hybrid model that merged Convolutional Neural Networks (CNNs) and Long Short-Term Memory (LSTM) networks. An improved model for multi-user cooperative spectrum sensing was created by using the capability of CNN to extract local features and the LSTM’s ability to process sequential data, thus leading to higher accuracy and efficiency in sensing. Moreover, a multi-head self-attention mechanism was added to smooth information exchange, thus making the model more adaptable and robust in dealing with fluctuating and intricate environments. The improved model produced low sensing error rates throughout different secondary user counts, with an extremely low sensing error under the setting of multiple users. Furthermore, the improved model had slow response times during sensing because of the complex sequential processing and the large number of parameters involved.

Table 1 represents the summary of the literature survey. From Table 1, ref. [16] lacked generalization under non-Gaussian noise, ref. [17] had limited portability due to hardware-specific calibration, ref. [18] had increased inference latency unsuitable for real-time applications, ref. [19] suffered from inconsistent performance under varying initializations, ref. [20] required large labeled datasets for training, posing real-world deployment challenges, ref. [21] risked selecting suboptimal detectors, ref. [22] limited its model evaluation to ResNet, overlooking hybrid architectures that could enhance performance, ref. [23] had high computational complexity, hindering real-time use, ref. [18] increased memory overhead, limiting deployment on edge devices, and ref. [25] faced synchronization issues in its parallel CNN–LSTM structure during training, affecting convergence stability. In [26], the model faced scalability issues when used in a large-scale dynamic CRN environment. In ref. [27], under low SNR, the model struggled to achieve high detection accuracy. The study in ref. [28] required a great deal of training data and high computational resources. In [29], small misconfiguration in the model degraded the detection accuracy. The study in ref. [8] involved an increase in latency due to complex processing. Therefore, to overcome these limitations, this research introduces a novel dual-layered Deep Q-Network architecture tailored to robust spectrum sensing in low-SNR environments. By integrating adaptive wavelet-based denoising and probabilistic time–frequency modeling into the DQN’s hidden layers, the proposed method enhances signal observability and decision confidence. It eliminates reliance on large labeled datasets, improves generalization under structured noise, and ensures real-time feasibility through compact noise-resilient feature representations.

In contrast to the existing research that treats these difficulties separately, the current research sets up a single framework that learns jointly from two aspects: the structured noise and the fragmented signal behavior through a reinforcement learning-driven adaptation, which constitutes simultaneous learning. The core contribution of this research is in bringing adaptive wavelet–cyclostationary denoising (ACWC-DAE) and probabilistic time–frequency reasoning (AGSTFT-DSM) within a Deep Q-Network framework for the first time. The integration contributes towards the sensing agent learning simultaneously from structured noise activities and uncertain spectral evidence, thus providing a different perspective on the signal observability restoration and uncertainty-aware decision fusion in very low-SNR cognitive radio environments. Such a fusion across domains of adaptive transforms, probabilistic modeling, and reinforcement learning has not previously been explored in the spectrum-sensing literature.

### Motivation

In cognitive radio networks, low-SNR conditions cause autocorrelation-based spectrum sensing to suffer from two intertwined degradation modes involving loss of signal observability and statistical distortion from structured noise. First, when signal power is extremely low, the inherent transitions in phase and amplitude states are suppressed by noise as the signal waveform becomes submerged within the noise envelope, effectively blurring distinct transitions and erasing temporal coherence. This leads to a collapse of the signal’s state-space dynamics and renders autocorrelation outputs statistically flat or chaotic. Simultaneously, unresolved sub-threshold transmissions and spectral leakage from adjacent activity introduce residual correlation within the noise itself, violating white Gaussian assumptions. These structured noise artifacts often exhibit lag-dependent behaviors that mimic weak primary user signals. The combined effect results in autocorrelation responses that either fail to converge or falsely peak, corrupting detection metrics and confusing signal presence with background distortion. This dual collapse of both signal structure and noise randomness poses a critical and underexplored challenge for reliable spectrum sensing in cognitive radio network deployments. Moreover, at low SNR, primary user signals that are either bursty or impacted by deep fading often appear fragmented and frequently dip below the noise floor. This occurs because instantaneous signal power fluctuates rapidly due to Rayleigh and Rician fading, causing the signal to periodically fall beneath the detection threshold. As a result, the signal is not continuously observable, and long stretches of the autocorrelation window contain missing or phase-incoherent segments. This disrupts the temporal consistency required for accurate autocorrelation since the correlation function relies on continuous or structurally coherent input over time. The resulting fragmentation prevents autocorrelation-based detectors from accumulating reliable correlation patterns, leading to missed detections even when the primary user is intermittently active. Conventional spectrum-sensing approaches fail to address this issue effectively, particularly in environments where low-duty-cycle transmissions dominate because the short and infrequent signal bursts offer limited observation time, making it difficult for fixed-window sensing methods to capture enough statistical evidence for reliable detection.

The existing spectrum sensing mechanisms, despite being improved with various advancements, are still not able to tackle the coexistence of the two degrading factors effectively, which are signal submergence in the noise envelope and structured noise correlation that violates the Gaussian assumption. The existing methods of energy, eigenvalue, and cyclostationary detection along with deep learning focus either on suppression or feature extraction but at the same time lose temporal coherence or cannot reason state under uncertainty from fragmented signal evidence. This unaddressed limitation constitutes the core research gap motivating this study. To bridge it, the proposed work will come up with a dual-layer Deep Q-Network that combines the Adaptive Continuous Wavelet Cyclostationary Denoising Autoencoder (ACWC-DAE) and the Adaptive Gaussian Short-Time Fourier Transform Dempster–Shafer Model (AGSTFT-DSM) to reconstruct weak temporal structures and carry out uncertainty-aware decision fusion in extremely low-SNR conditions. The detailed explanation for this proposed method is presented in the next section.

## 3. Deep Wavelet Cyclostationary Independent Gaussian Markov Fourier Transform Dempster–Shafer Network

To overcome the problems in sensing spectrum for detecting the primary user in low SNR, a novel “Deep Wavelet Cyclostationary Independent Gaussian Markov Fourier Transform Dempster–Shafer Network” is proposed. This proposed method is the combination of Adaptive Continuous Wavelet Cyclostationary Denoising Autoencoder (ACWC-DAE), Adaptive Gaussian Short-Time Fourier Transform Dempster–Shafer Model (AGSTFT-DSM), and Deep Q-Network (DQN). In the DQN, the ACWC-DAE is incorporated in the first hidden layer, and AGSTFT-DSM is incorporated in the second hidden layer.

Adaptive Continuous Wavelet Cyclostationary Denoising Autoencoder (ACWC-DAE)

This ACWC-DAE is the combination of Adaptive Continuous Wavelet Transform (ACWT), Cyclostationary Independent Component Analysis Detection (CICAD), and Denoising Autoencoder (DAE). This ACWC-DAE address the degradation modes of autocorrelation-based spectrum sensing in low-SNR cognitive radio networks. In this context, the spectrum-sensing task is formulated as a binary hypothesis testing problem, where the secondary user must decide whether the channel is occupied by a primary user or not based on noisy and incomplete signal observations. The contribution of each method is outlined below:Adaptive Continuous Wavelet Cyclostationary Denoising Autoencoder (ACWC-DAE)–The ACWT provides robust time–frequency decomposition that adaptively captures the transient and bursty features of weak primary user signals, even when temporal coherence is disrupted by noise.–This effectively restores the signal’s underlying structure by emphasizing energy concentrations across multiple scales.Cyclostationary Independent Component Analysis Detection (CICAD)–Simultaneously, the CICAD module exploits the inherent cyclostationary properties of modulated signals and performs blind separation of structured noise and spectral leakage artifacts using lag-dependent statistical independence.–It thereby disentangles primary user signal components from non-Gaussian correlated interference that violates white-noise assumptions.Denoising Autoencoder (DAE)–The DAE enhances this layered processing by learning nonlinear noise patterns and reconstructing clean signal representations, even when observability is severely degraded, allowing it to suppress residual structured noise while preserving useful signal characteristics.

Finally, by embedding these three complementary mechanisms within the DQN’s first layer, this approach enables the secondary user’s reinforcement learning agent to receive high-fidelity, denoised, and structure-aware signal representations. This empowers the DQN to learn robust sensing policies that are resilient to chaotic autocorrelation behavior, false peaks, and signal submergence, thus overcoming the core limitations of traditional autocorrelation-based detectors in low-SNR and spectrally crowded cognitive radio environments.

Adaptive Gaussian Short-Time Fourier Transform Dempster–Shafer Model (AGSTFT-DSM)

The AGSTFT-DSM effectively resolves the challenges posed by fragmented, bursty, and deep fading-affected primary user signals under low-SNR conditions. It is the combination of Adaptive Gaussian Mixture Hidden Markov Modeling (AGMHMM), Adaptive Short-Time Fourier Transform (ASFT), and Dempster–Shafer Theory (DST) into the second hidden layer of the Deep Q-Network (DQN). Within the binary hypothesis testing framework, this layer aids the secondary user in detecting intermittent primary user transmissions even when signal evidence is weak or partially obscured. The contributions made by each method are outlined below:Adaptive Gaussian Mixture Hidden Markov Modeling (AGMHMM)–The AGMHMM component models the underlying temporal dynamics of bursty signals by capturing probabilistic state transitions between active and inactive phases, even when signal fragments intermittently dip below the noise floor due to Rayleigh or Rician fading.–This enables the system to track hidden activity states across long sensing windows, where traditional autocorrelation methods fail.Adaptive Short-Time Fourier Transform (ASFT)–The ASFT provides adaptive resolution in the time–frequency domain, dynamically adjusting its windowing to resolve brief energy bursts and phase-incoherent segments, thereby recovering partial temporal structure that fixed-window detectors overlook.Dempster–Shafer Theory (DST)–The incorporation of DST enables robust evidence fusion by quantifying uncertainty and combining probabilistic beliefs from the HMM and ASFT modules, especially useful in low-duty-cycle environments where only weak and sporadic signal evidence is available.

Therefore, this combination allows the second layer of the DQN to construct a coherent and confidence-weighted internal representation of primary user signal presence, even when direct observation is unreliable or sparse. Ultimately, by integrating these adaptive, probabilistic, and uncertainty-aware mechanisms, this method enables accurate detection of fragmented and intermittently active primary user signals, overcoming the limitations of conventional autocorrelation and fixed-window spectrum-sensing techniques in severely degraded cognitive radio network scenarios.

Figure 1 illustrates the overall architecture of the proposed Deep Wavelet Cyclostationary Independent Gaussian Markov Fourier Transform Dempster–Shafer Network for robust spectrum sensing in cognitive radio networks under low-SNR conditions. The input signal is first passed through the Adaptive Continuous Wavelet Cyclostationary Denoising Autoencoder (ACWC-DAE), which integrates Adaptive Continuous Wavelet Transform for capturing bursty signal structures, Cyclostationary Independent Component Analysis Detection for separating structured noise, and Denoising Autoencoder for reconstructing clean signal representations. These enhanced features are fed into the first hidden layer of a Deep Q-Network, allowing it to learn reliable detection policies. The processed signal then flows into the second hidden layer, which hosts the Adaptive Gaussian Short-Time Fourier Transform Dempster–Shafer Model (AGSTFT-DSM). This layer combines Adaptive Gaussian Mixture Hidden Markov Modeling to model temporal activity states, Adaptive Short-Time Fourier Transform for dynamic time–frequency adaptation, and Dempster–Shafer Theory for uncertainty-aware fusion. Together, these modules allow accurate detection of intermittent or fragmented signals, and the final decision layer outputs the presence or absence of primary user activity.

### 3.1. Adaptive Continuous Wavelet Cyclostationary Denoising Autoencoder

In a cognitive radio network, the degradation modes of spectrum sensing in the low-SNR problem are solved using the novel Adaptive Continuous Wavelet Cyclostationary Denoising Autoencoder. To determine if the channel is being used by the primary user (PU), Equation (Equation 1) defines the signal that the secondary user (SU) receives [31](1)$${\left\{R{\left(n\right)}_{i}\right\}}_{n=1}^{N_{l}}=\left\{\begin{array}{cc} {\left\{W{\left(n\right)}_{i}\right\}}_{n=1}^{N_{l}}, & \mathrm{if}\;h_{0}\\ {\left\{GS{\left(n\right)}_{i}+W{\left(n\right)}_{i}\right\}}_{n=1}^{N_{l}}, & \mathrm{if}\;h_{1} \end{array}\right.$$

In this case, $N_{l}$ represents the received signal’s sample length, $R{\left(n\right)}_{i}$ represents the nth received signal sample during the i-th detection period, $W{\left(n\right)}_{i}$ represents Adaptive White Gaussian Noise with zero mean, *G* represents channel gain, and $S{\left(n\right)}_{i}$ represents the PU signal. This White Gaussian Noise accurately signifies the thermal noise component of the channel, and it allows for the controlled manipulation of the signal-to-noise ratio. It ensures the variations in the performance are mainly due to the sensing model rather than the unpredictable fading effects. Then, after, $h_{0}$ indicates that the PU is not occupying the channel, whereas $h_{1}$ indicates that the PU is occupied. If the channel is occupied, it is denoted as label 1, and, if not occupied, it shows label 0. The problem is that existing spectrum-sensing methods struggle to classify whether the channel is occupied or not. The proposed method begins by capturing the received IQ signal at the secondary user (SU). This IQ signal includes only noise (if the primary user (PU) is absent) or a mix of PU signal with noise (if the PU is present). This corresponds to the binary classification in spectrum sensing.

The input IQ signal is directly passed into the first hidden layer of a Deep Q-Network (DQN), which is enhanced with three complementary components: Adaptive Continuous Wavelet Transform (ACWT), Cyclostationary Independent Component Analysis Detection (CICAD), and Denoising Autoencoder (DAE). The ACWT adaptively analyzes the IQ signal across multiple time–frequency scales by correlating it with a mother wavelet at various scales (which correspond to different frequencies) and time shifts. This process generates a two-dimensional map, or scalogram, revealing how different frequencies of the signal change over time. Mathematically, the adaptive wavelet coefficients of the received signal are computed in Equation (Equation 2) [32]:(2)$$A_{R}\left(a,b\right)=\int_{-\infty }^{\infty }R{\left(t\right)}_{i}\;\cdot \;\psi_{a,b}^{*}\left(t\right)\;dt$$
where $R{\left(t\right)}_{i}$ is the time-domain representation of the received signal during the *i*-th detection interval, and $\psi_{a,b}^{*}\left(t\right)$ is the scaled and shifted version of the adaptive wavelet. The adaptivity is achieved by dynamically selecting the optimal wavelet function that minimizes a reconstruction-based loss or maximizes signal energy concentration, as defined in Equation (Equation 3):(3)$$\psi_{\mathrm{opt}}^{*}=\arg\min_{\begin{array}{c} \psi \in \Psi \end{array}}L_{\mathrm{recon}}\left(R{\left(t\right)}_{i},\psi \right)$$

The filter embedded in the Adaptive Continuous Wavelet Transform (ACWT) is an adaptive Gaussian-modulated band-pass filter that is functionally equivalent to a Morlet-type wavelet kernel. The Gaussian envelope provides smooth energy localization with the lowest side-lobe leakage, and the complex sinusoidal modulation allows fine frequency discrimination for the oscillatory signal components. In the time–frequency domain, this combination creates the band-pass response that satisfies the minimum-uncertainty principle, offering the finest joint resolution. The wavelet rapidly focuses on short high-frequency bursts or gradual low-frequency fluctuations because the adaptive Gaussian filter automatically adjusts its center frequency and bandwidth based on the signal’s immediate energy concentration. This ability to adapt is critical for CRN spectrum sensing, where the primary user signal is bursty, weak, or deeply faded. The adaptive Gaussian modulated filter used in the ACWE-DAE indicates lower reconstruction error and higher SNR recovery compared to existing wavelet families, thus proving potential for accurate signal recovery and feature extraction in low-SNR cognitive radio network environments.

The adaptivity in ACWT means that the choice of the wavelet function itself can be dynamically adjusted to best match the characteristics of the specific signal being analyzed at a given time or frequency scale. This is particularly valuable when dealing with signals containing complex and rapidly changing features as a standard CWT with a fixed mother wavelet might not be optimal for all parts of such a signal. By adapting the wavelet, ACWT can achieve a more precise and accurate representation of the signal’s local characteristics, leading to improved analysis and feature extraction. The core benefit of this adaptivity lies in its ability to effectively highlight transient bursts and localized structures within a signal, even in the presence of significant noise and temporal incoherence. Traditional methods like the Fourier Transform excel at analyzing signals with consistent frequency content but struggle with transient events or signals whose frequency characteristics change over time. ACWT overcomes this limitation by offering enhanced localization in both time and frequency, effectively acting like a variable zoom lens, focusing on short-duration high-frequency details and longer lower-frequency trends simultaneously. This makes it particularly well-suited for applications like identifying weak primary user (PU) signals in a noisy radio spectrum or detecting subtle anomalies in vibration signals from machinery, where the ability to pinpoint these elusive features amidst the background noise is crucial for accurate detection and analysis. Its robustness to noise and temporal variations in the signal makes ACWT a powerful tool for analyzing complex real-world data, where signal properties are often unpredictable and subject to interference. To complement the ACWT module, the Cyclostationary Independent Component Analysis Detection (CICAD) module exploits the inherent periodic statistical structure of modulated signals for robust feature extraction under severe interference. Cyclostationary signals possess periodically varying mean and autocorrelation functions, enabling detection based on spectral correlation functions (SCFs) rather than pure energy content. This allows signal identification even when energy levels fall below the noise floor. Mathematically, $R{\left(t\right)}_{i}$ is the cyclic autocorrelation function and is defined in Equation (Equation 4) [33]:(4)$$R_{\mathrm{cys}}^{\alpha \tau }=\lim_{T\to \infty }\frac{1}{T}\int_{-T}^{T}R{\left(t+\frac{\tau }{2}\right)}_{i}\;R^{*}{\left(t-\frac{\tau }{2}\right)}_{i}\;e^{-j2\pi \alpha t}\;dt$$

Here, α represents the cycle frequency, and τ is the delay variable. The SCF $S^{\alpha }\left(f\right)$, derived from the Fourier transform of $R_{cys}^{\alpha }\left(\tau \right)$, reveals the spectral redundancy induced by the modulating waveform. This property is not present in stationary noise, enabling discrimination between modulated signals and Gaussian interference. To separate the underlying primary user (PU) signal from correlated background interference, an Independent Component Analysis (ICA) is applied in the frequency domain. Let $X\;\epsilon \;R^{M\times N}$ represent the matrix of spectral correlation features, and then ICA decomposes as represented in Equation (Equation 5) [34]:(5)X=A•S
where *A* is the mixing matrix, and *S* is the matrix of statistically independent source signals. The demixing matrix *W* is learned such that S=W·X. This operation enables blind separation of primary user signal components from structured noise or spectral leakage artifacts that do not conform to cyclostationary assumptions.

To enhance the robustness of the extracted features, a Denoising Autoencoder (DAE) is employed as the final stage in the first hidden layer of the network. The DAE is trained to learn a mapping from noisy signal representations to their clean counterparts by minimizing reconstruction error. Given a corrupted version of the signal $\tilde{R}{\left(t\right)}_{i,}$, the encoder ε(·) projects it to a latent representation $Z_{i}$, and the decoder D(·) reconstructs the clean signal $\hat{R}{\left(t\right)}_{i}$. The process is outlined in Equation (Equation 6) [35]:(6)$$Z_{i}=\varepsilon \left(\tilde{R}{\left(t\right)}_{i},\;\hat{R}{\left(t\right)}_{i}\right)=D\left(Z_{i}\right)$$

The loss function for training the DAE is defined as in Equation (Equation 7):(7)$$L_{\mathrm{DAE}}=\frac{1}{N_{l}}\sum_{n=1}^{N_{l}}{\left(R{\left(n\right)}_{i}-\hat{R}{\left(n\right)}_{i}\right)}^{2}$$

This enables the network to learn nonlinear noise distributions and preserve salient structures of the original signal, enhancing detection sensitivity under degraded observability. Furthermore, by embedding Adaptive Continuous Wavelet Transform (ACWT), Cyclostationary Independent Component Analysis Detection (CICAD), and Denoising Autoencoder (DAE) within the first hidden layer of a Deep Q-Network (DQN), the agent receives a highly enriched representation of the environment’s spectral state. This improved state representation enables robust Q-value prediction for channel occupancy decisions even under chaotic SNR conditions, bursty noise, and false peaks, which typically hinder traditional autocorrelation-based detection schemes.

### 3.2. Adaptive Gaussian Short-Time Fourier Dempster–Shafer Layer

To further improve spectrum sensing accuracy under highly fragmented, intermittent, and fading-affected scenarios, an additional Adaptive Gaussian Short-Time Fourier Dempster–Shafer (AG-STFT-DS) layer is embedded in the second hidden layer of the Deep Q-Network. This layer integrates three complementary components: Adaptive Gaussian Mixture Hidden Markov Modeling (AGMHMM), Adaptive Short-Time Fourier Transform (ASFT), and Dempster–Shafer Theory (DST), to construct a confidence-weighted temporal and spectral inference module capable of reasoning under uncertainty and limited observations. The AGMHMM models the temporal evolution of primary user (PU) signal presence as a discrete-time Markov process with multiple hidden states representing transmission behavior (e.g., active, idle, or fuzzy). Given a sequence of received signal features $\left\{X_{1},\;X_{2},\ldots \ldots ,X_{T}\right\}$, the AGMHMM assumes that each observation $X_{t}$ is generated by a mixture of Gaussian emissions associated with a hidden state $z_{t}$. The observation likelihood is computed as in Equation (Equation 8) [36]:(8)$$p\left(\frac{X_{t}}{z_{t}}=c\right)=\sum_{k=1}^{K}\alpha_{c,k}\;\cdot \;N\left(X_{t};\;\mu_{c,k},\;\Sigma_{c,k}\right)$$
where $\alpha_{c,k}$ are the mixture weights, and $\mu_{c,k}$, $\Sigma_{c,k}$ are the mean and covariance of the *k*-th Gaussian component in state *c*. The parameters are iteratively estimated using the Expectation–Maximization (EM) algorithm to maximize the log-likelihood defined in Equation (Equation 9):(9)$$\theta^{*}=\arg\max_{\theta }\sum_{t=1}^{T}\log\;p\left(\frac{X_{t}}{\theta }\right)$$

This probabilistic modeling enables the detection of transient transmission bursts and deep fades where the primary user signal intermittently disappears below the noise floor.

Complementing this, the Adaptive Short-Time Fourier Transform (ASFT) provides localized spectral analysis that dynamically adjusts its window function to optimize resolution in time and frequency. For a received signal ${R(t)}_{i}$, the ASFT is computed as in Equation (Equation 10) [37]:(10)$$\mathrm{ASFT}\left(t,f\right)=\int_{-\infty }^{\infty }R{\left(t_{0}\right)}_{i}\;\cdot \;h_{\mathrm{opt}}\left(t_{0}-t\right)\;\cdot \;e^{-j2\pi ft_{0}}\;dt_{0}$$

Here, $h_{opt}\left(t\right)$ is an adaptive Gaussian window selected based on local signal energy concentration to balance trade-offs in time and frequency resolution. The adaptivity allows the system to effectively capture short-duration bursts and phase-incoherent segments that would otherwise be smeared by fixed-window STFTs. To consolidate evidence from the AGMHMM and ASFT components under uncertainty, Dempster–Shafer Theory (DST) is employed. DST formulates reasoning in terms of belief masses $m_{i}\left(A\right)$, where A⊆Θ and $\Theta =\left\{\mathrm{PU}\;\mathrm{absent},\mathrm{PU}\;\mathrm{present},\;\mathrm{fuzzy}\right\}$ is the frame of discernment. Each module contributes its belief assignment over Θ, and Dempster’s rule of combination fuses them as in Equation (Equation 11) [38]:(11)$$\begin{array}{c} \left(m_{1}⊕m_{2}\right)\left(A\right)=\frac{1}{1-K}\sum_{B\cap C=A}m_{1}\left(B\right)\;\cdot \;m_{2}\left(C\right)\\ K=\sum_{B\cap C=\emptyset }m_{1}\left(B\right)\;\cdot \;m_{2}\left(C\right) \end{array}$$

The belief Bel (*A*) and plausibility Pl (*A*) functions are then derived in Equations (Equation 12) and (Equation 13):(12)$$\mathrm{Bel}\left(A\right)=\sum_{B\subseteq A}m\left(B\right)$$(13)$$\mathrm{Pl}\left(A\right)=\sum_{B\cap A\neq \emptyset }m\left(B\right)$$

These metrics characterize the lower and upper bounds of the confidence in the PU’s state, allowing the DQN to reason under ambiguity and enforce conservative decisions in marginal cases.

By integrating AGMHMM, ASFT, and DST into the second hidden layer, the DQN builds a belief-aware, time–frequency adaptive, and temporally probabilistic representation of the signal. This enables robust spectrum sensing even when signals are deeply faded, fragmented, or in inconsistently observable conditions under which traditional energy detection or fixed-threshold methods fail. The reinforcement learning agent, guided by this structured internal representation, can dynamically optimize its sensing policy and reliably determine primary user presence under highly adverse channel conditions.

### 3.3. Novel Deep Q-Network Architecture

The spectrum-sensing task, as described in Section 3.1 and Section 3.2, is framed as a binary hypothesis classification problem under uncertainty:$H_{0}$: The channel is idle (primary user is absent);$H_{1}$: The channel is occupied (primary user is present).

The ultimate decision on whether the primary user (PU) is occupying the channel is made by a Deep Q-Network (DQN), which integrates the denoised structure-enhanced features from Section 3.1 and the temporally probabilistic and evidence-fused outputs from Section 3.2 to form a Q-value-based spectrum-sensing policy. The DQN consists of the following core layers:Input Layer: Raw IQ signal samples ${R(n)}_{i}$ captured at the secondary user (SU) over a time window of length $N_{l}$.First Hidden Layer: The Adaptive Continuous Wavelet Cyclostationary Denoising Autoencoder (Section 3.1) processes the signal to extract high-fidelity multiscale cyclostationary-aware denoised features $f^{1}\epsilon R^{d_{1}}$.Second Hidden Layer: The Adaptive Gaussian Short-Time Fourier Dempster–Shafer Layer (Section 3.2) models the temporal and frequency-domain uncertainty and structure through AGMHMM, ASFT, and DST, generating a feature vector $f^{2}\epsilon R^{d_{2}}$.

These two outputs are concatenated and passed to the fully connected Q-layer as defined in Equation (Equation 14):(14)$$f_{\mathrm{concat}}=\left[f^{\left(1\right)}\;∥\;f^{\left(2\right)}\right]\in R^{d_{1}+d_{2}}$$

This vector is then processed by a fully connected layer to produce Q-values for each possible action in Equation (Equation 15):(15)$$Q\left(s,a;\;\theta \right)=\mathrm{DQN}\left(f_{\mathrm{concat}};\;\theta \right)$$
where

*s*: current observation state of the spectrum (formed from the concatenated feature vector);a∈0,1: sensing action (0 = declare idle, 1 = declare busy);θ: parameters of the DQN (weights and biases).

The DQN is trained using Q-learning, with experience tuples ${(s}_{t},a_{t},\;r_{t},\;s_{t+1})$ stored in an experience replay buffer and updated iteratively based on the Bellman Equation (Equation 16):(16)$$Q\left(s_{t},a_{t}\right)\leftarrow Q\left(s_{t},a_{t}\right)+\alpha \left[r_{t}+\gamma \max_{a^{\prime }}Q\left(s_{t+1},a^{\prime }\right)-Q\left(s_{t},a_{t}\right)\right]$$
where α is the learning rate, γ is the discount factor, and $r_{t}$ is the reward based on the correct classification of primary user state at time *t*.

Reward Model

The reward $r_{t\;}$ is defined as follows:+1: Correct detection (PU correctly identified as present or absent);−1: Incorrect detection (false alarm or missed detection);−0.5: Uncertain or fuzzy detection with partial evidence.

This reward guides the DQN to learn a robust sensing policy, especially in low-SNR regimes, where signal components are fragmented, submerged, or corrupted by structured noise.

Final Output Decision

The final decision at the output layer of the DQN is a binary action based on the learned Q-values:(17)$$a^{*}=\arg\max_{a\in \{0,1\}}Q\left(s,a\right)$$
where

$a^{*}=1\;\Rightarrow \;$ primary user is present (channel is busy);$a^{*}=0\;\Rightarrow \;$ primary user is absent (channel is idle).

A binary decision framework $\left(h_{0},h_{1}\right)$ is used, with $h_{0}$ representing the situation where the primary user is absent (noise only) and $h_{1}$ indicating the primary user is present (signal and noise). This approach is very realistic for the case of spectrum access given that it all comes down to the secondary user’s decision, which will eventually determine whether a channel is free or occupied for transmission. The final output is indeed binary; however, the proposed ACWC-DAE and AGSTFT-DSM layers simulate internally intricate channel behaviors, such as those caused by partial data, fading, and overloaded or noisy conditions, by adaptively separating structured noise and reconstructing signal components before classification. In case the input only contains noise, the DAE department eliminates random and correlated noise patterns, thus producing a flat feature response that the DQN identifies as $h_{0}$. Additionally, the reinforcement-learning reward mechanism serves as an error self-correction process by penalizing false alarms and missed detections, which also helps in refining sensing policy over time. As a result, the binary decision output stays accurate and trustworthy even in mixed or non-ideal channel scenarios. This final output inherently integrates the multi-resolution wavelet features, cyclostationary components, temporal burst modeling, and uncertainty fusion, producing a trustworthy spectrum-sensing decision even under severely degraded signal environments. Overall, this research introduces a novel DRL-based spectrum-sensing framework tailored to cognitive radio networks operating under very low-SNR conditions. By integrating an Adaptive Continuous Wavelet Cyclostationary Denoising Autoencoder into the first hidden layer and an Adaptive Gaussian Short-Time Fourier Transform Dempster–Shafer Model into the second hidden layer of a Deep Q-Network (DQN), the system effectively captures, denoises, and interprets weak and fragmented primary user signals. The proposed approach addresses key limitations in conventional methods, such as autocorrelation collapse, signal submergence, and structured noise confusion. It enables robust binary hypothesis detection even when signal observations are incomplete or heavily distorted. The reinforcement learning agent is trained to develop optimal sensing strategies based on enhanced signal representations. As a result, the model significantly improves detection accuracy and reliability in spectrally crowded and fading-impacted environments.

## 4. Results and Discussion

The Results and Discussion section presents a detailed evaluation of the proposed Deep Wavelet Cyclostationary Independent Gaussian Markov Fourier Transform Dempster–Shafer Network under varying SNR conditions. Performance metrics such as detection probability, false alarm rate, sensing error, classification accuracy, and detection time are analyzed to validate the proposed model’s robustness. Comparative graphs illustrate the effectiveness of each integrated module across diverse signal environments. The results demonstrate significant improvements in accuracy, reliability, and responsiveness over traditional spectrum-sensing techniques in the CRN environment.

### 4.1. Dataset Description

This dataset is designed for research and development in cognitive radio networks (CRNs), specifically focusing on cluster-assisted spectrum sensing. The dataset includes a variety of features that represent signal characteristics, environmental conditions, and network parameters to facilitate spectrum sensing and decision-making. This dataset has been optimized for machine learning-based spectrum-sensing techniques, enabling efficient and dynamic spectrum utilization. It can be used for tasks such as primary user detection, spectrum availability prediction, and interference minimization.

The key features include signal strength parameters such as Received Signal Strength Indicator (RSSI), signal-to-noise ratio (SNR), and Power Spectral Density (PSD). Environmental factors include noise level, interference levels, and channel conditions. Network metrics cover cluster ID, sensing duration, and transmission success rate. Temporal features involve the time of spectrum sensing and the periodicity of spectrum availability. The target variable is either binary (0: Spectrum Unavailable, 1: Spectrum Available) or a multiclass label based on interference levels [39].

### 4.2. Hyperparameters

Several hyperparameters were set for the experimental setup to improve the model training and evaluation process. The Denoising Autoencoder (DAE) was trained for 50 epochs using a batch size of 32 with a dropout rate of 0.2, which is good for regularization and avoiding overfitting. The Q-learning component used a learning rate of 0.001 to have stable policy updates, whereas the Expectation–Maximization (EM) process was performed 100 times to ensure convergence. The dataset was divided in such a way that 70% was used for training and 30% for testing, which improved the model performance evaluation, which was both balanced and representative.

### 4.3. System Configuration

OS: Windows 11 64-bit operating system, ×64-based processor;RAM: 64.0 GB;Processor: Intel (R) Core(TM) i7-9700 CPU @ 3.00 GHz (3.00 GHz);Tool: Python version 3.14.0.

### 4.4. Simulation Results

This section presents the simulation results of the proposed spectrum-sensing framework. The structural architecture of the ACWC-DAE and AGSTFT-DSM modules is illustrated in Figure 2, Figure 3, Figure 4, Figure 5 and Figure 6, respectively.

Figure 2 shows the Power Spectral Density (PSD) comparison between “PU Active” and “PU Inactive” states across a frequency range of 0–500 Hz. The PU Active curve consistently shows slightly elevated PSD values compared to the PU Inactive curve, particularly in the 50–250 Hz range. This reflects the presence of structured energy patterns introduced by the primary user’s signal that are detectable despite low-SNR conditions. This result demonstrates the effectiveness of the proposed Deep Wavelet Cyclostationary Independent Gaussian Markov Fourier Transform Dempster–Shafer Network, captures energy bursts across time–frequency scales, distinguishes true signal components from correlated noise, and then reconstructs the clean signal. These modules enhance PSD clarity by restoring the signal’s spectral structure and suppressing non-Gaussian noise artifacts, allowing DQN to learn spectral patterns even under deep fading and bursty transmission conditions.

Figure 3 represents the Inter-Channel Spectral Coherence during PU transmission, showing how coherence magnitude varies with frequency from 0 to 500 Hz during primary user signal transmission. Coherence magnitude starts high, close to $10^{0}$, indicating strong inter-channel correlation at lower frequencies. Notable dips occur at around 90 Hz, 190 Hz, 370 Hz, and 480 Hz, where the coherence magnitude drops below $10^{-2}$, showing weak correlation and possible interference or noise artifacts. Despite these dips, most of the spectrum maintains a magnitude between $10^{-1}$ and 100, suggesting stable coherence in primary signal bands. These results confirm the presence of consistent signal patterns across channels, disrupted only at specific frequency notches. The data validates the method’s ability to isolate coherent structures even under fluctuating conditions. The results confirm the method’s efficiency in maintaining spectral integrity and improving detection accuracy.

Figure 4 illustrates the relationship between average SNR in decibels and the probability of detection for different primary user presence states. The SNR values range from 5 to 27 dB, while the detection probability ranges from 0.80 to 0.98. Blue and orange dots represent detection outcomes in the absence and presence of the primary user, respectively. As SNR increases, detection probability clusters between 0.93 and 0.98, confirming strong performance at higher signal strengths. Even at lower SNR values between 5 and 10 dB, the detection probability remains high, ranging from 0.82 to 0.90. This stability reflects the contribution of ACWC-DAE in signal restoration and noise suppression. The AGSTFT-DSM enhances detection by modeling bursty activity and combining uncertain observations for enhancing high detection accuracy.

Figure 5 shows primary user detection confidence over time, with the *y*-axis representing confidence scores from 0.5 to 1.0 and the *x*-axis covering a time index of 1000. The confidence score fluctuates but remains consistently above 0.5, indicating that the model maintains a strong baseline belief in detection reliability. Peaks reaching between 0.95 and 1.0 demonstrate high certainty during active primary user presence. Occasional dips suggest temporary uncertainty caused by noise or signal fading. The frequent oscillations reflect the system’s responsiveness to real-time changes in signal strength and structure. This dynamic behavior is driven by adaptive feature extraction across multiple layers. Overall, the high average detection confidence confirms the system’s effectiveness in monitoring intermittent primary user activity.

Figure 6 illustrates primary user signal reconstruction during active periods by comparing the noisy input signal with the reconstructed version over 400 samples. The raw waveform represents the raw input, fluctuating with high amplitude from 5 to 28 units, reflecting strong interference and noise. The reconstructed signal, with amplitude ranging from about −5 to +5, reveals a much cleaner version of the primary user transmission. This reconstruction is achieved by the DAE embedded in the framework, which effectively suppresses both structured and random noise while preserving essential signal features. The ACWT and CICAD modules help to isolate time–frequency and cyclostationary characteristics before processing. This validates its capability for spectrum sensing in low-SNR environments, ensuring reliable detection.

### 4.5. Performance Evaluation

The performance metrics of the proposed model for spectrum sensing under low-SNR conditions, along with the enhanced detection accuracy and reduced sensing error achieved through the ACWC-DAE and AGSTFT-DSM mechanisms, are explained in detail in this section.

Figure 7 illustrates how well the system identifies the presence of a primary user under varying signal-to-noise-ratio conditions. The SNR value is −20 dB and the detection probability is 0.55, indicating moderate sensitivity in extreme noise. The SNR is −5 dB and detection reaches 0.92, showing a steep improvement in accuracy. At SNR is 5 dB, the curve saturates close to 0.98, reflecting near-perfect detection. This significant improvement is achieved by the Adaptive Gaussian Short-Time Fourier Transform Dempster–Shafer Model, which tracks fragmented and fading signals and fuses uncertain evidence to reinforce detection reliability. The method excels in resolving bursty or intermittent signals hidden below the noise floor, making it ideal for enhancing detection performance under adverse conditions.

Figure 8 shows the relationship between probability of detection and correlation coefficient (ρ). The ρ is 0.2 and the probability of detection achieves 0.81, indicating good performance even under weak correlation conditions, which often represent fragmented or noise-blurred signals. At ρ is 0.6, the detection probability improves to 0.91, showing growing confidence as signal coherence increases. Finally, when ρ is 1.0, the detection rate is 0.98, indicating perfect detection under fully coherent signals. The Adaptive Continuous Wavelet Cyclostationary Denoising Autoencoder suppresses structured lag-dependent noise and restores the temporal and cyclostationary structure of weak signals, which directly boosts detection as ρ increases, and it ensures robustness against chaotic autocorrelation behavior, enabling more accurate detection as correlation improves.

Figure 9 illustrates the probability of false alarm vs. probability of detection, a classic ROC curve evaluating the trade-off between true and false detections. When the detection rate is 0.2, the false alarm probability is 0.58, suggesting conservative detection with minimal false alarms. As the probability of detection increases to 0.6, the probability of false alarm rises to 0.9, showing increasing susceptibility to false positives. Finally, the detection is 1.0 and the probability of false alarm reaches 0.98, indicating aggressive detection settings where noise and structured interference are misidentified as signals. The Adaptive Continuous Wavelet Cyclostationary Denoising Autoencoder minimizes false alarms by disentangling structured noise artifacts from true signals using cyclostationary separation and nonlinear denoising. Its significance lies in preserving good signal structure while suppressing false peaks caused by autocorrelation collapse, thus achieving a better balance in detection performance under noisy conditions.

Figure 10 shows the variations in sensing error vs. SNR, where sensing error represents the rate of incorrect detection decisions. The SNR is −20 dB, sensing error is 0.09, and the SNR is −5 dB; the error drops to 0.06, indicating improved accuracy as the signal becomes more discernible. When SNR is 5 dB, the curve flattens near 0.05, suggesting stabilized and reliable performance. The Adaptive Gaussian Short-Time Fourier Transform Dempster–Shafer Model models bursty signals, dynamically adjusting to signal length and strength, with uncertain evidence to reduce misclassifications. This layered and adaptive strategy enables robust detection despite deep fading and signal fragmentation, thus minimizing sensing errors, especially under challenging SNR conditions.

Figure 11 illustrates the variations in detection time regarding different SNR levels. As the SNR increases from −20 dB to +5 dB, the detection time consistently decreases, indicating improved performance in cleaner signal conditions. At −20 dB, the detection time is 51 ms, which gradually reduces to 45 ms at −15 dB, 40 ms at −10 dB, and 35.5 ms at −5 dB. At 0 dB, the time drops further to 32 ms, reaching its minimum of 30.10 ms at +5 dB. This trend confirms that higher SNR enables faster and more reliable signal detection. The Adaptive Gaussian Short-Time Fourier Transform Dempster–Shafer Model (AGSTFT-DSM) framework effectively enhances time–frequency resolution and enables quicker responses even in noisy environments. The method’s ability to reduce detection latency while preserving accuracy showcases its significance in spectrum-sensing applications. Overall, the AGSTFT-DSM outperforms conventional models by adapting efficiently to varying noise conditions.

### 4.6. Comparison Evaluation

In this section, to emphasize the significance of the proposed work, a comparative analysis is carried out in terms of probability of detection, correlation coefficient, detection time, classification accuracy, and sensing error against several conventional techniques, including Convolutional Long Short-Term Deep Neural Network (CLDNN), Convolutional Long Short-Term Memory Network (ConvLSTM) [40], energy detection (ED), Correlation Matrix-Based Convolutional Neural Network (CM-CNN), Attention-Based Primary User Activity Spectrum Sensing (APASS), Dempster–Shafer based Multi-Model Adaptive Spectrum Sensing Algorithm (DS2MA) [41], Hierarchical Cooperative LSTM, Convolutional Neural Network–Long Short-Term Memory Network (CNN–LSTM), APASS [42], Convolutional Neural Network (CNN) [24], Deep Neural Network [24], and Long Short Term Memory [24].

Figure 12 presents the probability of detection (%) for three different models: CLDNN, ConvLSTM, and the proposed method. The probability of false alarm is 1.0. At the time the CLDNN model achieves the lowest detection probability at 78%, ConvLSTM model improves significantly with a detection probability of 96%. The proposed method achieves the highest detection probability of 97.8%, clearly outperforming the others. The proposed method reaches near-perfect detection, ConvLSTM slightly lower, and CLDNN considerably behind. This comparison confirms that the proposed method achieved the best detection performance, while CLDNN performed the poorest among the three methods.

Figure 13 illustrates the correlation coefficient (ρ) performance across five spectrum-sensing models under low-SNR conditions. The traditional energy detection (ED) method shows the lowest correlation with a value of 0.20. The CM-CNN model moderately improves this, achieving 0.60. APASS and DS2MA demonstrate high correlation values of 0.97 and 0.98, respectively, reflecting their strong detection performance. However, the proposed method achieves the highest correlation coefficient of 0.99, indicating good actual primary user activity. This validates its effectiveness in preserving structural signal features and reducing false detection under severe noise conditions in CRN.

Figure 14 shows the detection time (ms) across the models, including CM-CNN, APASS, DS2MA, and the proposed method. The APASS model exhibits the highest detection time at 37.17 ms, indicating slower performance. The DS2MA model follows with 34.19 ms, and CM-CNN performs slightly better at 31.95 ms. The proposed method demonstrates the lowest detection time of 30.10 ms, indicating the fastest signal detection capability among all the models. The proposed method is the most efficient and APASS the slowest. This analysis confirms that the proposed method is the fastest, while APASS has the highest delay in detection.

Figure 15 presents a comprehensive comparison of classification accuracy across varying signal-to-noise-ratio (SNR) levels, ranging from −20 dB to −10 dB. At the SNR of −20 dB, the proposed method achieves the highest classification accuracy of 60%, outperforming the Hierarchical Cooperative LSTM at 57%, CNN–LSTM at 56%, and APASS at 55%. At −16 dB and −14 dB, the proposed method continues with good accuracy of 70% and 81%, respectively, while the other models exhibit slower gains. In −10 dB, the proposed method reaches 94% accuracy, greater than the Hierarchical Cooperative LSTM, CNN–LSTM, and APASS at 90%, 78%, and 70%. The consistent dominance of the proposed method across all noise levels showcases its enhanced feature extraction, robust noise suppression, and effective decision-making capabilities in low-SNR environments.

Figure 16 presents the sensing-error comparison across SNR levels (−20 dB to 5 dB) for CLDNN, ConvLSTM, and the proposed method. At −15 dB, the proposed method reaches 0.09, the CLDNN suffers from a high sensing error of 0.45, while ConvLSTM is 0.13. At the SNR of −5 dB, the proposed method significantly reduces its sensing error of 0.07 compared to ConvLSTM at 0.08 and CLDNN at 0.09, indicating strong resilience to noise. At 5 dB, all models converge to lower sensing errors, but the proposed method achieves the lowest at −0.05, confirming its superior ability to minimize misdetection across both low- and high-SNR environments.

Figure 17 illustrates the detection probability across the baseline models, including CNN, DNN, LSTM, and the proposed method. The SNR value is −20 dB. The LSTM method shows the lowest detection, with a value of 0.25; the CNN model moderately improves this, achieving 0.28. DNN shows the detection probability of 0.48, and the proposed method achieves a detection probability of 0.55. This indicates that the proposed method has a higher probability of detection in low SNR at −20 dB compared to the baseline models, clearly indicating that the proposed model achieved the best detection performance.

### 4.7. Comparison of the Proposed Model with Different Datasets

In this comparison section, two more datasets are used to train and test the proposed model. Dataset 1 [39] (already used in the paper), dataset 2 [43], and dataset 3 [44] are used to test the model efficiency, including parameters such as probability of detection and sensing error. This section proves that the proposed model operates efficiently under different datasets regarding the comparison of the parameters under −20 dB. The comparison graphs are presented below.

Figure 18 demonstrates the comparison of the detection probability of the proposed model across various datasets. At a low SNR of −20 dB, the proposed model achieves the highest probability of detection of 0.55 for dataset 1. Then, the proposed model shows small variations in values for probability of detection values of 0.54 and 0.53 for dataset 2 and dataset 3. Thus, the proposed model is not only suitable for dataset 1 but also dataset 2 and dataset 3, demonstrating that it efficiently provides results across various datasets.

Figure 19 exhibits a comparison of the sensing error across various datasets with an SNR value of −20 dB. The proposed model reaches the lowest sensing error of 0.09 for dataset 1. For dataset 2 and dataset 3, the sensing error of the proposed model reaches 0.10 and 0.11. This graph clearly shows that the overall error values are between 0.09 and 0.11; this variation shows that the proposed model consistently maintains efficiency and sensing accuracy under various types of datasets. This graph validates the consistency of the proposed method while it adapts under varying environmental conditions.

Overall, the performance evaluation demonstrates that the proposed method significantly outperforms the existing models across all the key spectrum-sensing metrics in low-SNR cognitive radio environments. It achieves the highest detection probability of 97.8%, the strongest correlation with actual signal activity of 0.99, and the lowest detection time of 30.10 ms, indicating rapid and accurate decision-making. Despite initially lagging in ROC performance under extremely low false alarm rates, the method becomes highly competitive as thresholds relax. Furthermore, its classification accuracy remains superior across a wide SNR range, achieving an impressive 94% accuracy at –10 dB, which exceeds all the existing models. Moreover, the proposed model is compared with baseline models and different types of datasets under −20 dB. In this scenario, the proposed method also performs more efficiently than other existing methods. This consistent advantage highlights the model’s powerful hybrid architecture, which effectively combines denoising, adaptive signal decomposition, and probabilistic reasoning to ensure high fidelity in spectrum sensing across diverse and challenging conditions.

### 4.8. Statistical Tests

In this section, statistical tests are conducted, and the test results are provided in Table 2 below.

Table 2 represents the statistical tests of the proposed model. This analysis demonstrates the consistent performance of the proposed model and high detection probability along important performance metrics. The mean of the probability of detection is 0.9587, the median is 0.9623, and the standard deviation is 0.0218, confirming that the proposed model consistently detects across multiple trials. These values demonstrate that the proposed model is statistically significant, with a *p*-value of 4.2 × $10^{-4}$ and a confidence interval in the range of 0.9525–0.9649. Similarly, the false alarm probability is 0.0413, which confirms that the proposed model clearly distinguishes between true and false detections. The sensing error has a mean of 0.0720, which is also supported by a small confidence interval (0.0649–0.0791). Therefore, the proposed model is accurate in its decision-making. The average detection time of 30.10 ms with a standard deviation of 5.00 ms indicates that the system has a rapid response capability, which is suitable for real-time CRN applications. All the *p*-values are significantly less than 0.01, which proves statistical significance and high reliability. Thus, the overall results are that the proposed method has high accuracy, a very low rate of false alarms, low sensing error, and high detection speed, thus confirming its strength and efficiency in low-SNR cognitive radio environments.

### 4.9. Ablation Study

This section describes the contribution of individual components within a proposed model by systematically removing them and observing the impact on performance.

Table 3 demonstrates the ablation study of the proposed model at −20 dB. Removal of one module from the whole proposed model leads to large decreases in overall performance regarding detection probability, sensing error, classification accuracy, and detection time. When ACWT is not included, the probability of detection decreases to 0.45, whereas the elimination of CICAD results in even lower detection of 0.40, along with a sensing error of 0.16, making it clear that cyclostationary separation is still needed for noise suppression. The removal of the Denoising Autoencoder (DAE) pushes the sensing error to 0.12, thereby proving its necessity for proper signal reconstruction. The neglect of the whole ACWC-DAE layer displays the greatest downfall in detection probability to 0.35 and classification accuracy to 45%. Likewise, the exclusion of AGMHMM, ASFT, or DST layers leads to weaker temporal tracking and evidence fusion. On the contrary, the complete removal of AGSTFT-DSM layer results in 0.30 detection probability and the highest sensing error of 0.22. Without the DQN decision layer, the performance drops to a detection probability of 0.28, and classification accuracy is 38%. Finally, the fully integrated model yields the best performance in probability of detection of 0.55, sensing error of 0.09, classification accuracy of 60%, and the lowest detection time of 51 ms. Hence, the cooperation of ACWC-DAE, AGSTFT-DSM, and DQN is proved, and the proposed model efficiently performs in a low-SNR environment.

## 5. Conclusions

In the proposed work, a novel low-SNR spectrum-sensing framework, the Deep Wavelet Cyclostationary Independent Gaussian Markov Fourier Transform Dempster–Shafer Network, was developed to enhance the robustness, accuracy, and responsiveness of cognitive radio systems. The framework integrates an Adaptive Continuous Wavelet Cyclostationary Denoising Autoencoder (ACWC-DAE) in the first layer and the Adaptive Gaussian Short-Time Fourier Transform Dempster–Shafer Model (AGSTFT-DSM) in the second layer of a Deep Q-Network. The experimental results show that the classification accuracy reaches 94% at SNR 5 dB while maintaining 60% even at −20 dB, validating strong resilience under severe noise. The detection probability improves from 0.55 at −15 dB to nearly 97.8 at 5 dB, and the sensing error reduces from 0.3 to 0.05 across the same range, confirming accurate and reliable detection. The detection time drops steadily from 51 ms at −20 dB to 30.10 ms at 5 dB. A correlation-based analysis shows high detection efficiency of 0.99. The ROC curve confirms low false alarm at low detection thresholds and controlled growth at higher thresholds due to ACWC-DAE’s false peak suppression. Overall, the proposed model successfully addresses chaotic autocorrelation behavior, bursty and fading signals, and structured noise artifacts, offering substantial improvements over traditional spectrum-sensing methods in noisy and dynamic radio environments.

### 5.1. Key Findings and Novel Insights

This research provides new insight into how degradation of spectrum sensing at low SNRs occurs, including the suppression of the signal within the noise and the correlation of structured noise, which breaks the Gaussian assumption. The proposed dual-layer DQN structure indicates that adaptive wavelet denoising followed by uncertainty-aware evidence fusion can lead to detection even under −20 dB SNR conditions. The combination of ACWC-DAE and AGSTFT-DSM indicates that a reinforcement learning agent has the ability to detect reliable sensing policies from partially observable and noisy signals, thus achieving a remarkable 0.55 probability of detection along with very few false alarms. This research opens a new path for deep reinforcement learning-based spectrum sensing by revealing the combined use of adaptive denoising and probabilistic reasoning to be a very robust solution in spectrally crowded and fading environments.

### 5.2. Future Work

For future work, the proposed framework can be extended to handle cooperative multi-user spectrum-sensing scenarios where multiple secondary users share partial observations. Integration with reconfigurable intelligent surfaces (RISs) can further enhance signal reconstruction under extreme fading. Lightweight variants could be developed for edge deployment in IoT applications. Additionally, exploring meta-learning strategies allows rapid adaptation to new signal environments. Cross-layer optimization can also be considered to align sensing with higher-layer communication protocols.

## Figures and Tables

**Figure 1 sensors-25-07361-f001:**
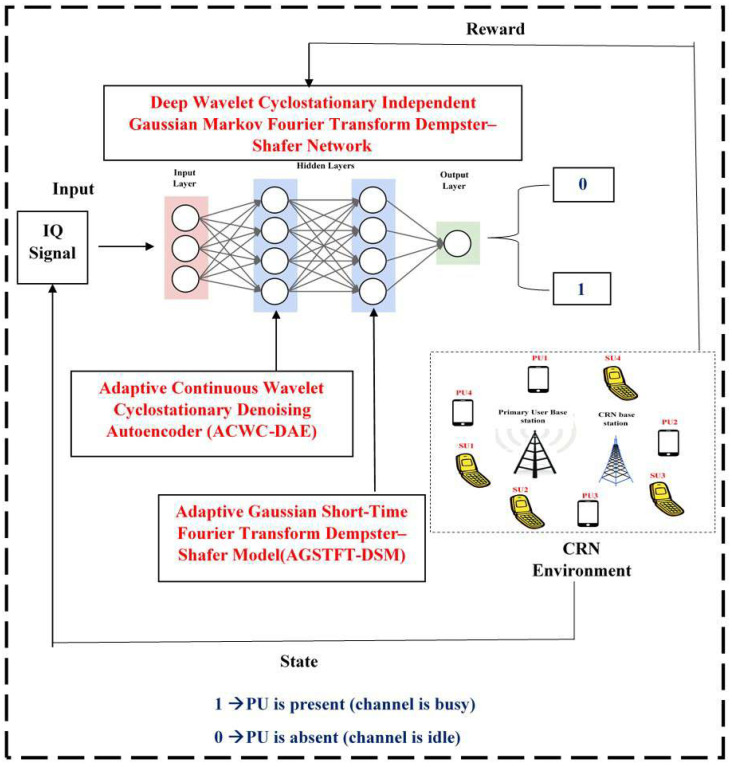
Architecture diagram for Deep Wavelet Cyclostationary Independent Gaussian Markov Fourier Transform Dempster–Shafer Network.

**Figure 2 sensors-25-07361-f002:**
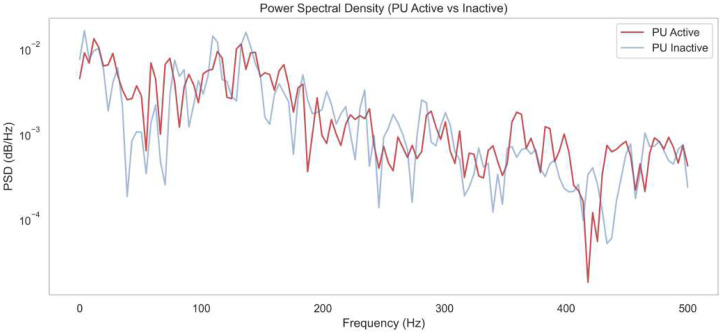
Power Spectral Density.

**Figure 3 sensors-25-07361-f003:**
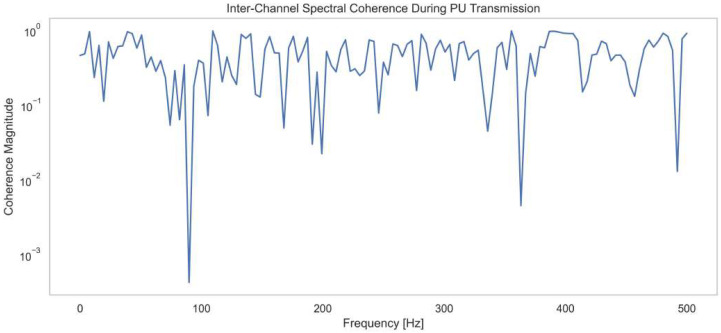
Inter-Channel Spectral Coherence during primary user (PU) transmission.

**Figure 4 sensors-25-07361-f004:**
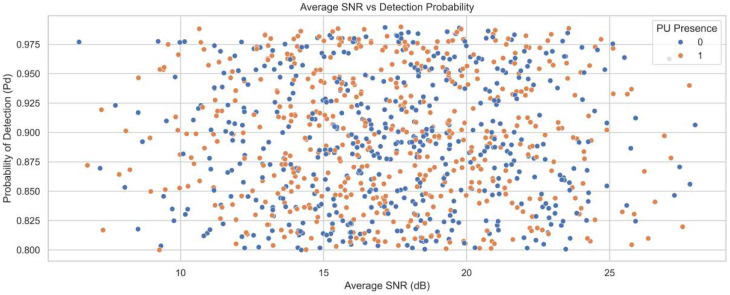
Average SNR (dB) and the probability of detection.

**Figure 5 sensors-25-07361-f005:**
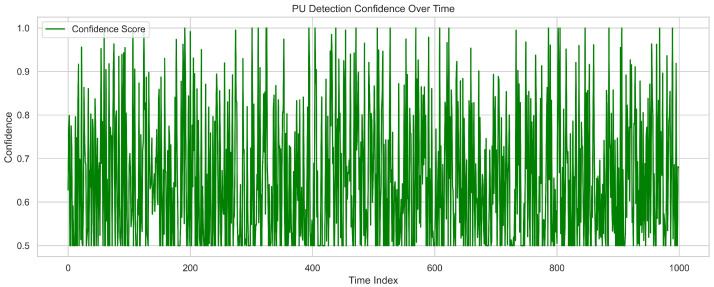
PU detection confidence over time.

**Figure 6 sensors-25-07361-f006:**
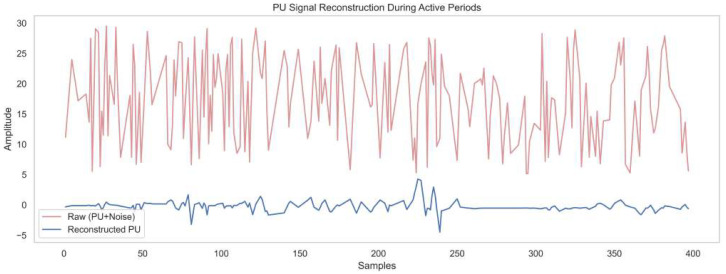
PU signal reconstruction.

**Figure 7 sensors-25-07361-f007:**
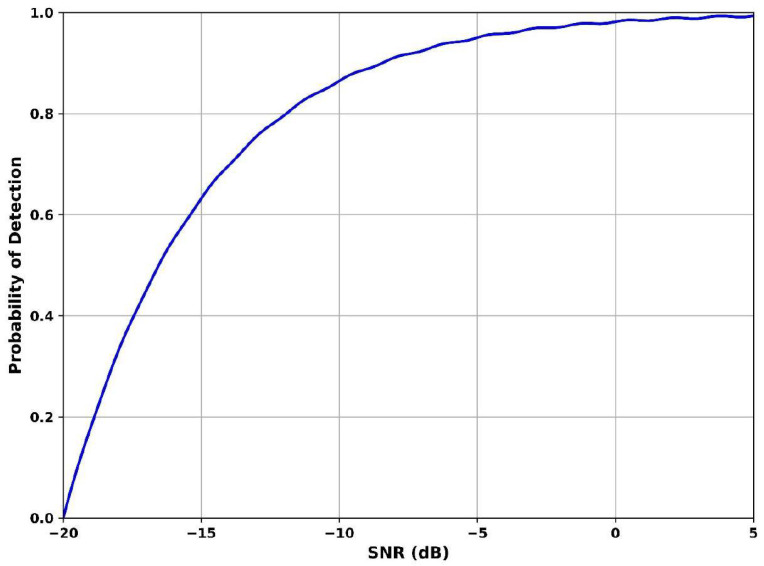
Probability of detection versus SNR.

**Figure 8 sensors-25-07361-f008:**
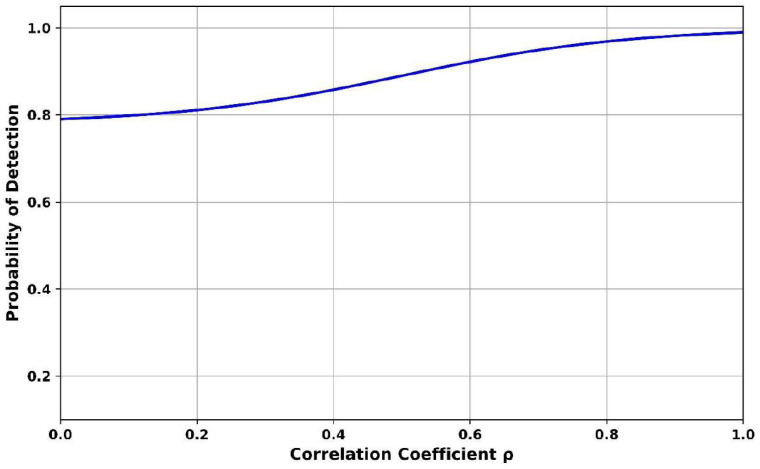
Probability of detection versus correlation coefficient.

**Figure 9 sensors-25-07361-f009:**
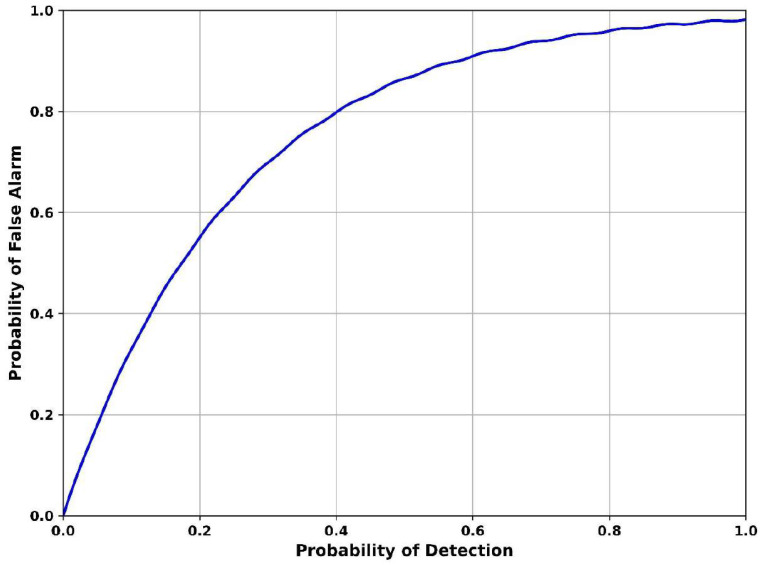
Probability of false alarm versus probability of detection.

**Figure 10 sensors-25-07361-f010:**
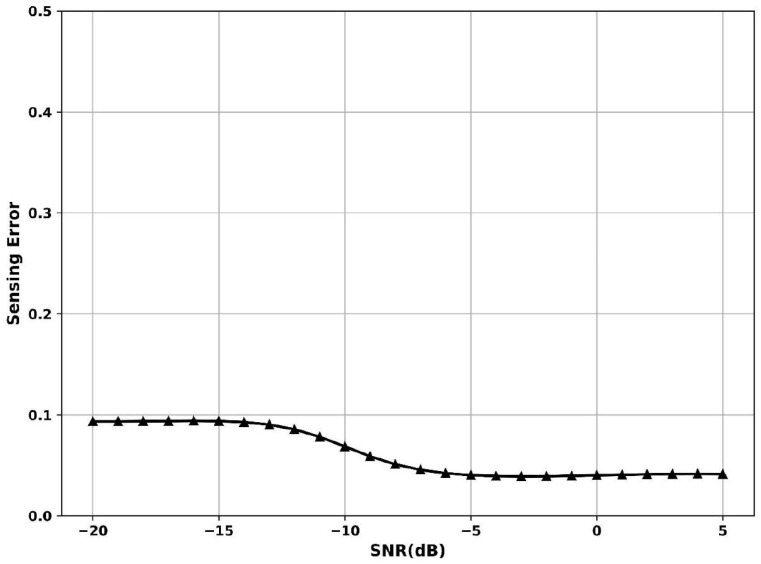
Sensing error versus SNR.

**Figure 11 sensors-25-07361-f011:**
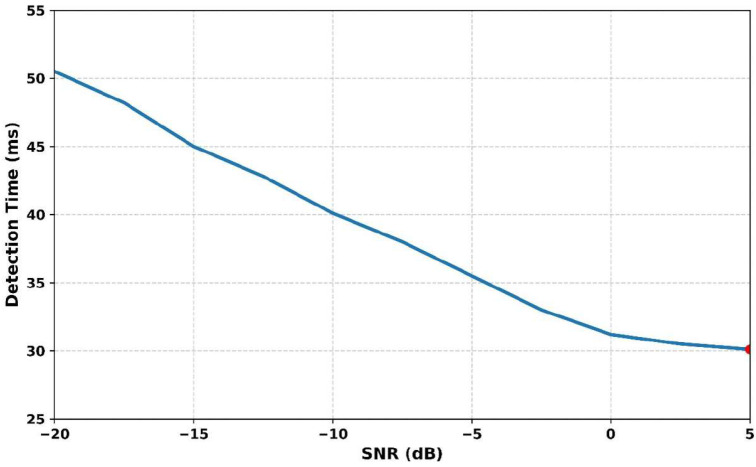
Detection time.

**Figure 12 sensors-25-07361-f012:**
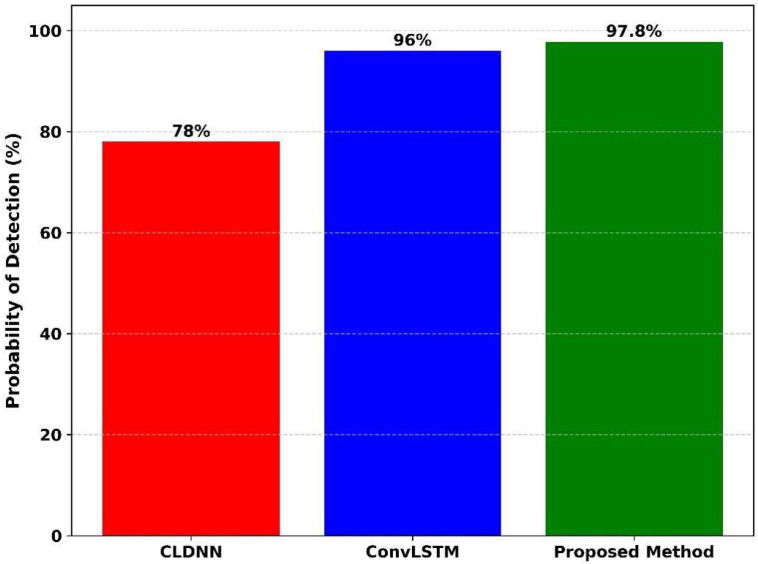
Comparison of probability detection.

**Figure 13 sensors-25-07361-f013:**
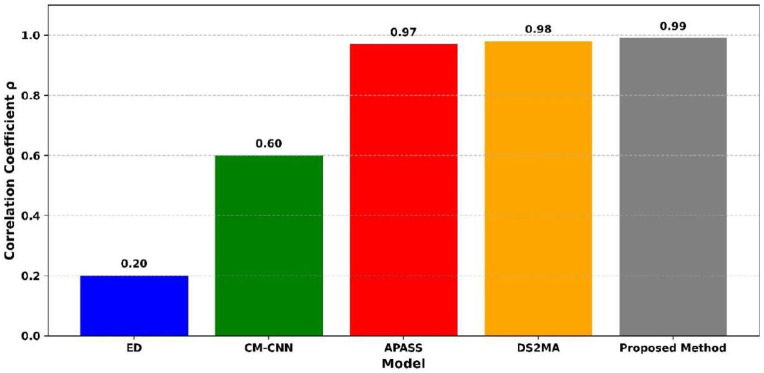
Comparison of correlation coefficients.

**Figure 14 sensors-25-07361-f014:**
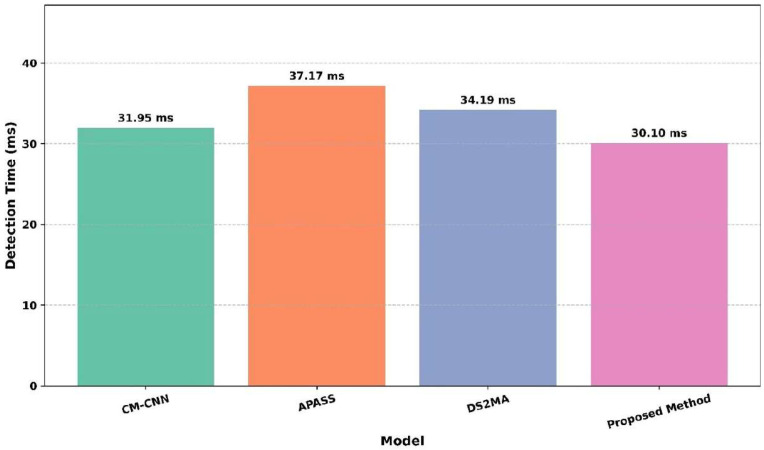
Comparison of detection time.

**Figure 15 sensors-25-07361-f015:**
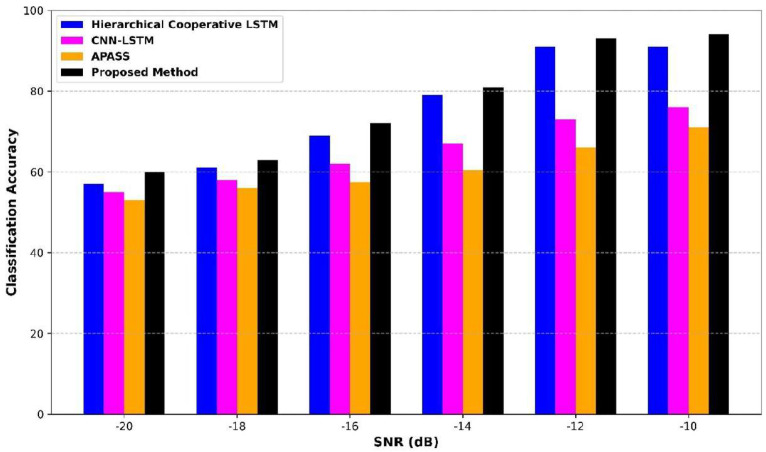
Comparison of classification accuracy.

**Figure 16 sensors-25-07361-f016:**
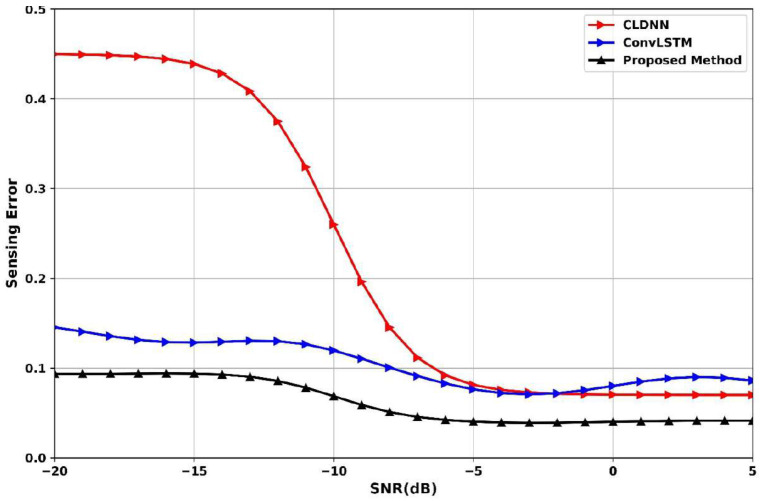
Comparison of sensing error.

**Figure 17 sensors-25-07361-f017:**
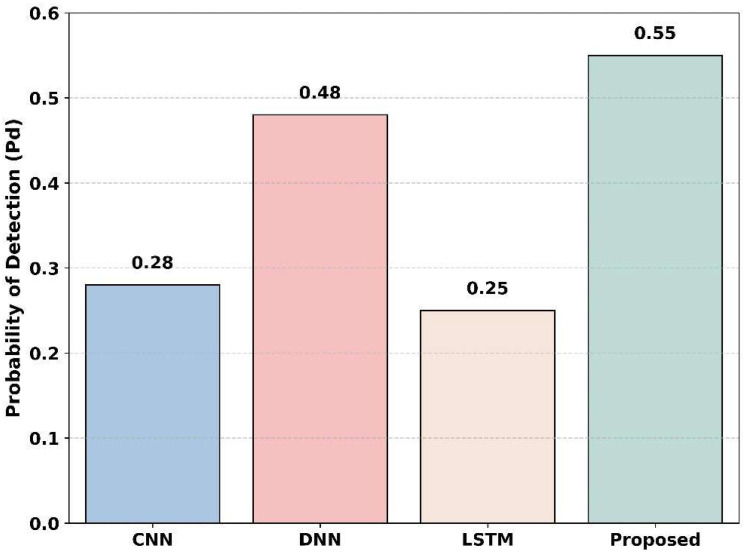
Comparison of probability of detection under −20 dB SNR.

**Figure 18 sensors-25-07361-f018:**
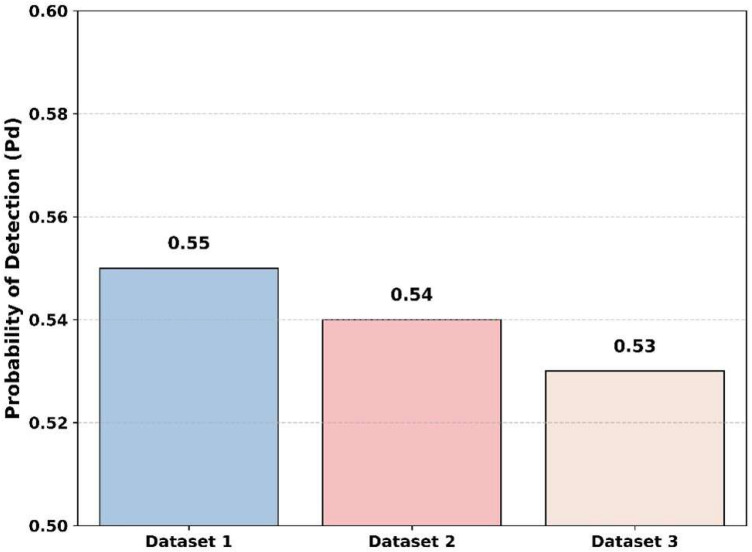
Comparison of probability of detection across various datasets.

**Figure 19 sensors-25-07361-f019:**
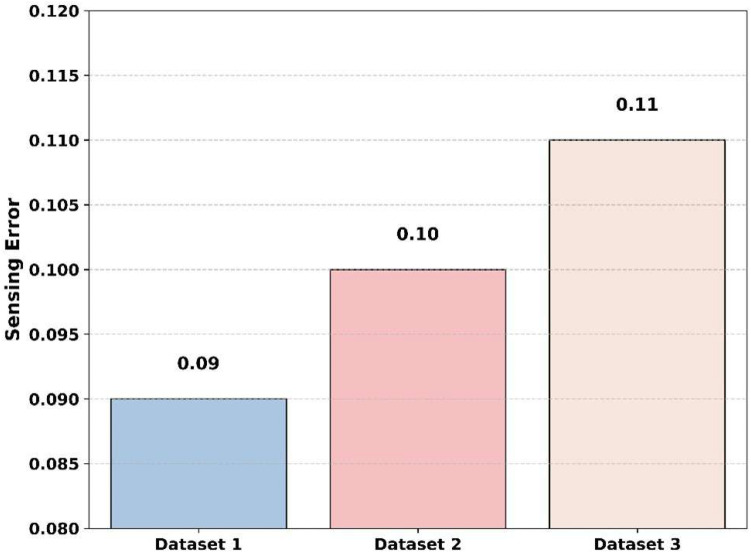
Comparison of sensing error across various datasets.

**Table 1 sensors-25-07361-t001:** Summary of the literature survey.

Ref.	Model	Significance	Limitation
[16]	CNN-based spectrum sensing	Identifies unused bands using CNN classification, improving accuracy and efficiency.	Weak generalization under non- Gaussian conditions.
[17]	DRAIN-NETS	Learns temporal and environmental features, enhancing sensing performance.	Limited portability across devices; requires accurate calibration.
[18]	DLSenseNet	Captures temporal dependencies and spatial relations from structural sensing data.	Increased latency due to high model complexity.
[19]	Adversarial learning-based	Extracts SNR-independent features for efficient sensing.	Uneven performance under different initialization setups.
[20]	DLbD	Uses LSTM to extract features from low-SNR modulated signals.	Requires large labeled datasets for effective training.
[21]	Multi-stage detector	CNN-based signal estimation improves detection accuracy via optimized weights.	Depends on preset thresholds, risking incorrect detector choice.
[22]	ResNet-based detection	Automatically categorizes data from time-series analysis.	Detection efficiency reduced due to baseline limitations.
[23]	CNN–TN	Captures both local and global signal features.	Single-model design increases complexity; hybrids raise computational cost.
[24]	CNN–RNN	Learns spatio-temporal features for strong regularization and generalization.	Increases memory overhead.
[25]	Parallel CNN–LSTM	Prevents information loss from serial structures, enhancing sensing reliability.	Synchronization issues during training affect stability.
[26]	DBN–CSA SST–CRN	Achieves nonlinear thresholding and distinguishes complex spectral patterns.	Lacks scalability for large dynamic CRN environments.
[27]	SVM–ensemble sensing	Enables distributed energy detection for cooperative sensing.	Accuracy decreases under low SNR.
[28]	DRL–CNN fused	Extracts RF spectrogram features representing signal density variations.	High data and resource needs delay real-time adaptation.
[29]	Fast Slepian Transform (FST)	Efficiently maps signals to sparse spectrum for accurate sub-band detection.	Minor misconfigurations severely impact accuracy.
[30]	CNN–LSTM with Attention	Combines local feature extraction and sequential modeling to reduce sensing errors.	Complex sequential processing slows model response.

**Table 2 sensors-25-07361-t002:** Statistical tests for the proposed method.

Metric	Mean	Median	Std. Dev.	*p*-Value	Confidence Interval
Probability of Detection	0.9587	0.9623	0.0218	$4.2\times 10^{-4}$	0.9525–0.9649
Probability of False Alarm	0.0413	0.0396	0.0139	$3.9\times 10^{-3}$	0.0374–0.0453
Sensing Error	0.0720	0.0700	0.0250	$2.5\times 10^{-3}$	0.0649–0.0791
Detection Time	30.10	30.10	5.00	$1.2\times 10^{-3}$	28.68–31.52

**Table 3 sensors-25-07361-t003:** Ablation study for the proposed method.

Configuration	Pd	Sensing Error	Accuracy (%)	Time (ms)
Without Adaptive CWT	0.45	0.14	53	58
Without Cyclic ICA Detection	0.40	0.16	50	60
Without Denoising Autoencoder	0.48	0.12	55	55
Without ACWC-DAE	0.35	0.20	45	65
Without Adaptive GMM-HMM	0.42	0.15	52	57
Without Adaptive STFT	0.44	0.13	54	56
Without DST Fusion	0.38	0.18	48	62
Without AGSTFT-DSM	0.30	0.22	40	70
Without DQN (Static Detector)	0.28	0.25	38	75
Full Model (ACWC-DAE + AGSTFT-DSM + DQN)	0.55	0.09	60	51

## Data Availability

The data are publicly available.

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
