# Peer review of "Sensors2025, 25(23), 7361;https://doi.org/10.3390/s25237361"

_sensors, 2025, doi:10.3390/s25237361_

Round 1

Reviewer 1 Report

Comments and Suggestions for Authors

Figure 1 does not match the description given in the text. The way it is explained does not allow the reader to visualize the connection or interaction between the blocks, making it difficult for other researchers to reproduce the experiment and the results. Check lines from 311 – 331. 
What type of channel was considered for analysis?. Define or mention in a way that allows a better understanding of your work.
 In cognitive radio systems and wireless systems in general, they use a certain type of modulation/channel access. In his work these are not mentioned. If they were considered,  as affecting the signal detection process with low signal-to-noise ratio?

Author Response

We appreciate your timely and meticulous attention to our work.  We much appreciate your constructive feedback and the opportunity to enhance our manuscript.  Here are our comprehensive responses to your remarks.  A distinct file is appended for the revision responses.

Reviewer 2 Report

Comments and Suggestions for Authors

Review for the paper title: “Efficient Deep Wavelet Gaussian Markov Dempster–Shafer Network-Based Spectrum Sensing at Very Low SNR in Cognitive Radio Networks”

The paper proposed a novel Deep Wavelet Cyclostationary Independent Gaussian Markov Fourier Transform Dempster–Shafer Network. The authors should consider the following comments to improve the paper:

  • The paper contains a lot of abbreviations, which make the text difficult to understand, especially in the “contribution of research” section.
  • The literature review section is very limited. There is a limited number of papers presented. Authors should consider more studies, as well as adding a summary table at the end of the literature review section to present the state of the art.
  • Section 3 should be rewritten such that it will be easy to follow for the reader, “lines 264 to 317”; I do suggest bullet points.  
  • Section 4.1, make a proper citation for the dataset source instead of putting the link
  • What is the difference between experimental and simulation results in sections 4.2 and 4.3?
  • Minor issues needed to be fixed such as:
  • Funding statement needed to be added.
  • All highlighted words in yellow needed to be removed.
  • What is the type of Noise used in the channel? Explain and justify your selection.
  • What is the kind of filter embedder within the wavelets? Justify your design approach.
  • The authors proposed a binary decision approach that is based on (0,1). Is this realistic? it assumes either the channel is busy or it is ideal, but in reality, there is something different, i.e., (channel busy, channel overloaded, channel free, channel contains data, channel contains data and noise, channel with noise only …etc.). How the proposed approach can treat the noise signal only? Are there any types of error detection and correction approaches used?

Author Response

(The authors gave the same response as above.)

Reviewer 3 Report

Comments and Suggestions for Authors

The manuscript presents a deep reinforcement learning–based spectrum sensing framework for cognitive radio networks operating under very low signal-to-noise ratio (SNR) conditions. The proposed method integrates two key modules: ACWC-DAE for denoising and restoring weak signal structures and AGSTFT-DSM for modeling bursty and uncertain signal patterns. The approach aims to improve detection accuracy, reduce false alarms, and shorten detection time compared to conventional and deep learning–based sensing techniques. Experimental results demonstrate strong performance. However, some issues and concerns need to be addressed. Suggestions are below.

  1. Include more quantitative comparisons with baseline methods using standardized datasets or benchmarks. Add statistical tests (e.g., confidence intervals and p-values) to assess the significance of reported performance gains.
  2. Provide hyperparameters, training epochs, and dataset partitioning details to allow replication.
  3. Consider adding ablation studies to show the contribution of each module (ACWC-DAE, AGSTFT-DSM).
  4. Minor grammatical and stylistic edits are needed to improve the technical flow.

Author Response

(The authors gave the same response as above.)

Reviewer 4 Report

Comments and Suggestions for Authors

This work investigated Deep wavelet Gaussian Markov Dempster-Shafer network-based spectrum sensing in cognitive radio networks. The following comments are provided for consideration:

1. Motivation and Contribution

- The motivation and contributions of the paper are not sufficiently articulated. The authors are encouraged to more clearly highlight the research gap being addressed and specify the unique contributions of this work.

2. Research Challenges

- The paper does not clearly describe the specific challenges encountered in conducting this study relative to prior works. Please elaborate on the distinctive technical difficulties or limitations in existing research that the proposed approach aims to overcome.

3. Novel Insights

- The manuscript should explicitly state the novel insights or key findings that distinguish this work from related studies.

Author Response

(The authors gave the same response as above.)

Round 2

Reviewer 1 Report

Comments and Suggestions for Authors

Do you consider that there are significant differences in performance between ConvLSTM and the proposed method as shown in Figure 12? Similarly, consider the methods shown in Figure 13 regarding correlation coefficient?

On the other hand, what other advantage could the proposed method present?

Comments on the Quality of English Language

In general the writing is correct and the style allows for easy reading. However, I consider that a general review of the manuscript would be advisable in order to eliminate any type of error as much as possible. 

Author Response

(The authors gave the same response as above.)

Reviewer 4 Report

Comments and Suggestions for Authors

I have no further comments. This paper could be published as is.

Author Response

(The authors gave the same response as above.)
